# Risk stratification for the prediction of skeletal-related events in patients with castration-resistant prostate cancer with bone metastases

Masanori Hamada[1], Eiji Nakata[2]*, Ryuichi Nakahara[2], Shinsuke Sugihara[3], Haruyoshi Katayama[2], Takuto Itano[2], Tomohiro Inoue[2], Shota Takihira[2], Yoshiteru Akezaki[4], Toshifumi Ozaki[2]

1 Division of Physical Medicine and Rehabilitation, Okayama University Hospital, Okayama City, Okayama, Japan, 2 Department of Orthopedic Surgery, Okayama University Hospital, Okayama City, Okayama, Japan, 3 Department of Orthopedic Surgery, Shikoku Cancer Center, Matsuyama City, Ehime, Japan, 4 Division of Physical Therapy, Kochi Professional University of Rehabilitation, Tosa, Kochi, Japan

* eijinakata8522@yahoo.co.jp

## Abstract

Skeletal-related events (SREs) are common in patients with bone metastases from castration-resistant prostate cancer (CRPC). Despite advances in prostate cancer treatment, clinically validated predictive models for SREs in CRPC patients with bone metastases remain elusive. This gap in prognostic tools hinders optimal patient management and treatment planning for this high-risk population. This study aimed to develop a prediction model for SRE by investigating potential risk factors and classifying them into different groups. This model can be used to identify patients at high risk of SREs who need close follow-up. Between 2004 and 2013, 68 male patients with bone metastases from CRPC who were treated at our institute were evaluated for survival without SREs and survival without SREs of the spinal cord. The study analyzed clinical data at enrollment to identify risk factors for initial and spinal SREs. Multivariate analysis revealed that a high count of metastatic vertebrae, along with visceral or lymph node metastases, were significant risk factors. Patients were categorized into four subgroups based on the number of vertebral metastases and presence of visceral or lymph node metastases: 1) extensive vertebral and both types of metastases, 2) extensive vertebral without additional metastases, 3) some vertebral with other metastases, 4) some vertebral without additional metastases. The first SRE and spinal SRE occurred significantly sooner in the first subgroup compared to others. Incidence rates at 12 months for the first SRE were 56%, 40%, 27%, and 5%, and for the first spinal SRE were 47%, 40%, 27%, and 0% respectively. Patients with extensive vertebral and additional metastases require vigilant monitoring to mitigate SREs.

**Data availability statement:** We have deposited the anonymized dataset on figshare. The associated DOI is https://doi.org/10.6084/m9.figshare.29146070.

**Funding:** The author(s) received no specific funding for this work.

**Competing interests:** The authors have declared that no competing interests exist.

## Introduction

Bone metastases are common in patients with castration-resistant prostate cancer (CRPC), occurring in more than 90% of patients with metastatic CRPC [1–3]. Small bone metastases are asymptomatic but progress slowly, often leading to skeletal-related events (SREs). Recent advances in the systemic treatment of CRPC have increased the median survival of patients with CRPC [4,5]. As a result, the management of SREs has become more important due to the longer time it takes for tumors to develop bone metastases. SREs can include malignant spinal cord compression (MSCC), pathological fractures, radiotherapy (RT), or bone surgery [5–8]. Approximately 42–59% of CRPC patients with bone metastases experience SREs [2,6–8]. Moreover, bone-modifying drugs such as zoledronic acid and denosumab are widely used to prevent SREs, but some patients experience severe SREs, including fractures and paraplegia [6,7]. These complications lead to a significant decline in activities of daily living (ADL) and reduced quality of life [6,7]. If the onset of an SRE can be predicted, bone metastases can be monitored more effectively, potentially preventing or delaying pathological fractures and paraplegia due to MSCC. Recent research has explored the factors contributing to SREs in CRPC patients with bone metastases [2,7,8]. These include elevated serum alkaline phosphatase (ALP) levels, high urinary N-telopeptide cross-link/creatinine ratio, bone pain, visceral metastasis at diagnosis, and short progression to CRPC [2,7,8]. Establishing a predictive model for SREs would aid in incorporating this data into clinical practice. Despite the prevalence of bone metastases in CRPC patients, a reliable and comprehensive framework for predicting the likelihood of SREs has yet to be established. We investigated potential risk factors for SRE and developed a prediction model by combining these factors and stratifying them into different groups. The purpose of this risk stratification model was to identify high-risk SRE patients requiring diligent oversight to safeguard their ADLs.

## Materials and methods

This study was approved by the Research Ethics Committee of the Shikoku Cancer Center (approval number 2017-26), and was conducted according to the World Medical Association's Declaration of Helsinki. Due to the study's retrospective nature, the need for informed consent was waived. The medical records of patients diagnosed with bone metastases of CRPC and treated at our institute between May 2004 and December 2013 were retrospectively evaluated. The last follow-up was performed in March 2015. Patients with previously undetected bone metastases and SREs were excluded. Thus, a total of 68 patients were included. The median age was 72.5 years (range 55–90 years). The median follow-up was 23 months (3–95 months). During this period, 33 patients died from tumor progression, all with a median follow-up of 23 months (range, 3–67 months). The median follow-up for survivors was also 23 months (range, 3–95 months). Diagnostic imaging included computed tomography (CT) scans, spinal magnetic resonance imaging (MRI), and bone scintigraphy. For bone metastases assessment, especially vertebral involvement, each affected bone or vertebra was considered a single metastatic site, irrespective of lesion count within

it. Thirty patients presented with bone-only lesions. Others had metastases in the lung (n = 1); lymph nodes (n = 27); both lymph nodes and lung (n = 5); lymph nodes and liver (n = 4); and lymph node, liver, and lung (n = 1). All patients underwent hormone therapy, with 39 receiving zoledronic acid and 14 denosumab. The study evaluated the incidence of SREs post-enrollment and the risk factors for initial SREs, defined as pathological fractures, spinal cord compression, RT, or bone surgery.

Time to first SRE was defined as the date of enrolment to the date of the first SRE reported during follow-up. Patients who did not experience SREs were censored at last contact or death. Time to first spinal SRE was assessed in the same way. For overall survival (OS), death from any cause was considered an event. SRE-free survival and OS were assessed using the Kaplan-Meier method and compared statistically using the log-rank test. To assess risk factors for the first SRE and spinal SRE, clinical data at enrollment were evaluated. These included age, visceral or lymph node metastases, number of bone metastases, number of vertebral metastases (cervical, thoracic, lumbar), ALP, lactate dehydrogenase, and prostate-specific antigen levels at the time of CRPC diagnosis. Univariate analyses utilized Fisher's exact test, while multivariate analyses employed logistic regression. An association was considered significant if the p-value was less than 0.05. Results were analyzed using Bell Curve for Excel (Social Research and Information Corporation, Tokyo, Japan).

## Results

### First SRE and first spinal SRE after registration

In total, 30/68 (44%) patients developed their first SRE post-registration. The most common sites were the spine (27 patients), ribs (2 patients), and pelvis (1 patient). The types of SREs included vertebral fractures (n = 24), MSCC (n = 3), and pathological fractures (n = 3). Of the 24 spinal SREs, 21 were vertebral fractures and three were MSCCs. Radiotherapy was required for 24 patients. The median time to the first SRE was 11 months (range, 0.1–63). SRE-free survival rates at 6, 12, and 24 months were 88%, 74%, and 63%, respectively (Fig 1a). The median time to the first spinal SRE was also 11 months (range, 0.1–63). Spinal SRE-free survival rates at 6, 12, and 24 months were 88%, 77%, and 66%, respectively (Fig 1b).

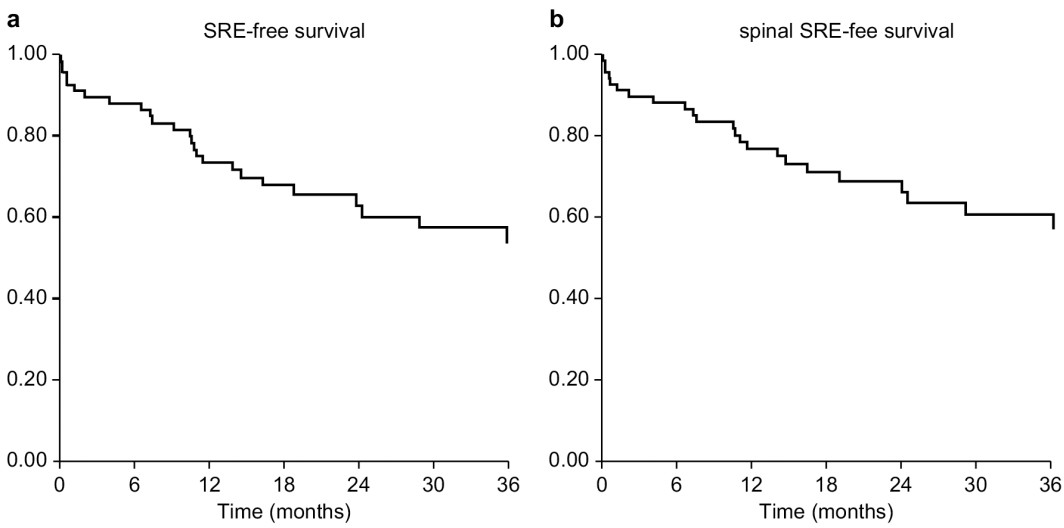

**Fig 1. SRE-free survival and spinal SRE-free survival. The SRE-free survival rates at 6, 12, and 24 months were 88%, 74%, and 63%, respectively (Fig 1a).** The spinal SRE-free survival rates at 6, 12, and 24 months were 88%, 77%, and 66%, respectively (Fig 1b). SRE, skeletal-related event.

## Risk factors for the first SRE development

Univariate analysis showed that extensive vertebral metastases (≥20) and visceral or lymph node metastases were risk factors for the first SRE (Table 1). Multivariate analysis identified two significant risk factors for the occurrence of the first SRE: extensive vertebral metastases (≥20) and the presence of visceral or lymph node metastases. Patients with extensive vertebral metastases showed a hazard ratio (HR) of 3.60 (95% CI: 1.09–11.9, p = 0.036), while those with visceral or lymph node metastases had an HR of 4.50 (95% CI: 1.44–14.03, p = 0.010).

The timing of the first SRE differed significantly between patient groups. For those with extensive vertebral metastases, the incidence of first SRE was markedly higher compared to patients with fewer than 20 vertebral metastases: 27% vs. 6% at 6 months and 51% vs. 17% at 12 months, respectively (Fig 2a). Similarly, patients with visceral or lymph node metastases experienced earlier onset of the first SRE compared to those without such metastases: 17% vs. 4% at 6 months, and 36% vs. 12% at 12 months, respectively (Fig 2b).

Univariate analysis showed that extensive vertebral metastases (≥20) and visceral or lymph node metastases were risk factors for the first spinal SRE (Table 2).

Multivariate analysis revealed two significant risk factors for the occurrence of the first spinal SRE: extensive vertebral metastases (≥20) and the presence of visceral or lymph node metastases. Patients with extensive vertebral metastases showed an HR of 3.44 (95% CI: 1.02–11.59; p = 0.046), while those with visceral or lymph node metastases had an HR of 6.81 (95% CI: 1.93–24.05; p = 0.003). Patients with extensive vertebral metastases (≥20) experienced their first spinal SRE considerably earlier than did those with fewer metastases. At 6 months, the incidence rates were 27% and 7%

**Table 1. Risk factors for first SRE development.**

| Covariates | Patients, (n) | | P-value |
| --- | --- | --- | --- |
| | Patients without first SRE | Patients with first SRE | |
| Age (years) | | | |
| <70 | 12 | 15 | 0.10 |
| ≥70 | 26 | 15 | |
| Visceral or lymph node metastases | | | |
| No | 21 | 6 | 0.003 |
| Yes | 17 | 24 | |
| Number of bone metastases | | | |
| <3 | 15 | 23 | 0.40 |
| ≥3 | 10 | 20 | |
| Number of vertebral metastases | | | |
| <20 | 32 | 17 | 0.012 |
| ≥20 | 6 | 13 | |
| ALP | | | |
| <300 | 17 | 13 | 0.052 |
| ≥300 | 21 | 17 | |
| LDH | | | |
| <200 | 18 | 13 | 0.47 |
| ≥200 | 20 | 17 | |
| PSA | | | |
| <10 | 26 | 14 | 0.06 |
| ≥10 | 12 | 16 | |

ALP, alkaline phosphatase; LDH, lactate dehydrogenase; PSA, prostate-specific antigen; SRE, skeletal-related event.

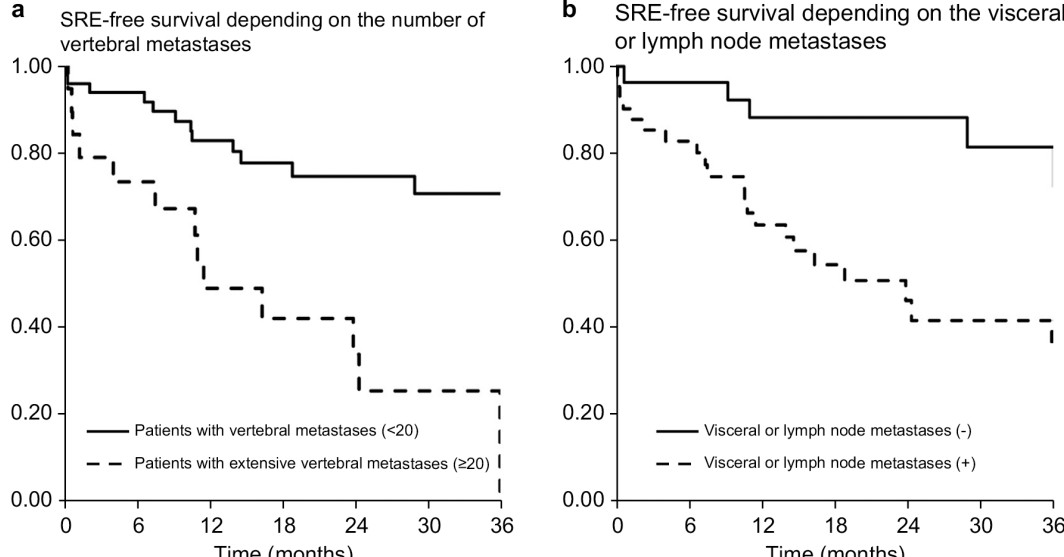

**a**  SRE-free survival depending on the number of vertebral metastases

**b**  SRE-free survival depending on the visceral or lymph node metastases

**Fig 2.  SRE-free survival depends on the number of vertebral metastases and visceral or lymph node metastases. Patients with extensive vertebral metastases (≥20) developed their first SRE significantly earlier than did those with <20 vertebral metastases; the SRE-free survival rates were 73% and 94% at 6 months and 49% and 83% at 12 months, respectively (Fig 2a).** Patients with visceral or lymph node metastases developed the first SRE significantly earlier than did those without visceral or lymph node metastases; the SRE-free survival rates were 96% and 83% at 6 months and 88% and 64% at 12 months, respectively (Fig 2b). SRE, skeletal-related event.

**Table 2.  Risk factors for first spinal SRE development.**

| Covariates | Patients, (n) | | P-value |
| --- | --- | --- | --- |
| | Patients without first spinal SREs | Patients with first spinal SREs | |
| Age (years) | | | |
| <70 | 14 | 13 | 0.18 |
| ≥70 | 27 | 14 | |
| Visceral or lymph node metastases | | | |
| Yes | 23 | 4 | <0.001 |
| No | 18 | 23 | |
| Number of bone metastases | | | |
| <3 | 17 | 8 | 0.23 |
| ≥3 | 24 | 19 | |
| Number of vertebral metastases | | | |
| <20 | 34 | 15 | 0.015 |
| ≥20 | 7 | 12 | |
| ALP | | | |
| <300 | 18 | 12 | 0.058 |
| ≥300 | 23 | 15 | |
| LDH | | | |
| <200 | 20 | 11 | 0.34 |
| ≥200 | 21 | 16 | |
| PSA | | | |
| <10 | 22 | 11 | 0.21 |
| ≥10 | 19 | 16 | |

ALP, alkaline phosphatase; LDH, lactate dehydrogenase; PSA, prostate-specific antigen; SRE, skeletal-related event.

for extensive and less extensive metastases, respectively, increasing to 45% and 15% at 12 months (Fig 3a). Similarly, patients with visceral and lymph node metastases developed their first spinal SRE notably earlier than those without. The incidence rates for patients with and without visceral metastases were 17% and 4% at 6 months, rising to 34% and 8% at 12 months (Fig 3b).

Patients were divided into four subgroups based on vertebral metastases and presence of visceral or lymph node metastases: 1) patients with ≥20 vertebral metastases and visceral or lymph node metastases; 2) patients with ≥20 vertebral metastases and no visceral or lymph node metastases; 3) patients with <20 vertebral metastases and visceral or lymph node metastases; 4) patients with <20 vertebral metastases and no visceral or lymph node metastases. Those in the first subgroup developed the first SRE and the first spinal SRE significantly earlier than other groups. The occurrence of initial SREs varied significantly across the subgroups over time. At 6 months, the incidence rates were 29%, 20%, 11%, and 0% for the respective subgroups. These rates increased by 12 months, reaching 56%, 40%, 27%, and 5% (Fig 4a). For spinal SREs, a similar trend was observed. The 6-month incidence rates were 30%, 20%, 11%, and 0%, increasing to 47%, 40%, 27%, and 0% by 12 months (Fig 4b). This data highlights the varying risks of SREs among different patient subgroups, with some groups experiencing a higher incidence of events, particularly as time progressed.

### Overall survival

Overall survival was 91% at 6 months and 88% at 12 months (Fig 5).

### Discussion

SREs were documented in 44% of CRPC patients with bone metastases in this study. The vertebral region accounted for 80% of the first SREs, making the spine the predominant site of occurrence. Patients with widespread vertebral metastases (≥20) or metastases in visceral or lymph nodes experienced their first SRE more rapidly. Additionally, initial spinal

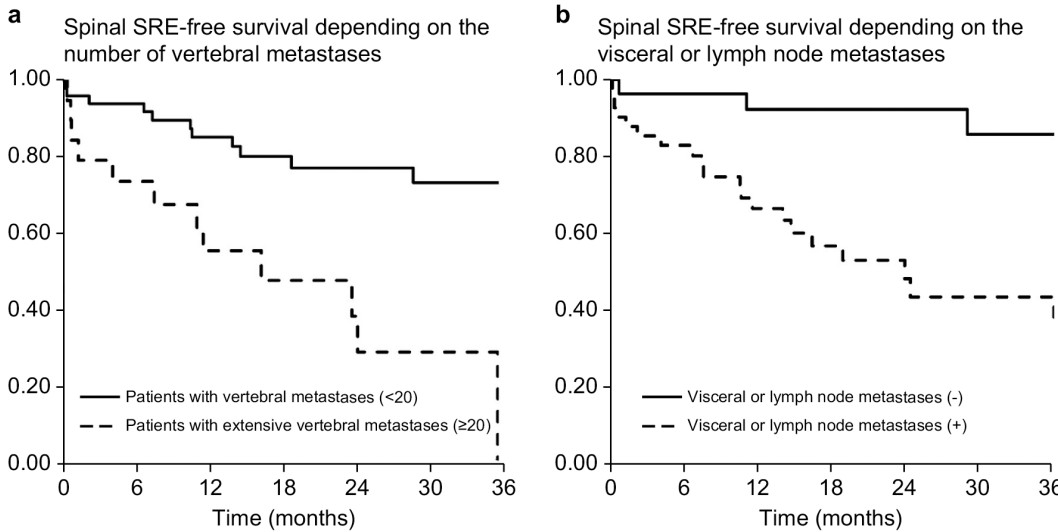

**Fig 3. Spinal SRE-free survival by the number of vertebral metastases and visceral or lymph node metastases.** Patients with extensive vertebral metastases (≥20) developed the first spinal SRE significantly earlier than did those with <20 vertebral metastases; the spinal SRE-free survival rates were 73% and 93% at 6 months and 55% and 85% at 12 months, respectively (Fig 3a). Patients with visceral or lymph node metastases developed the first spinal SRE significantly earlier than those without visceral metastases; the spinal SRE-free survival rates were 83% and 96% at 6 months and 66% and 82% at 12 months, respectively (Fig 3b). SRE, skeletal-related event.

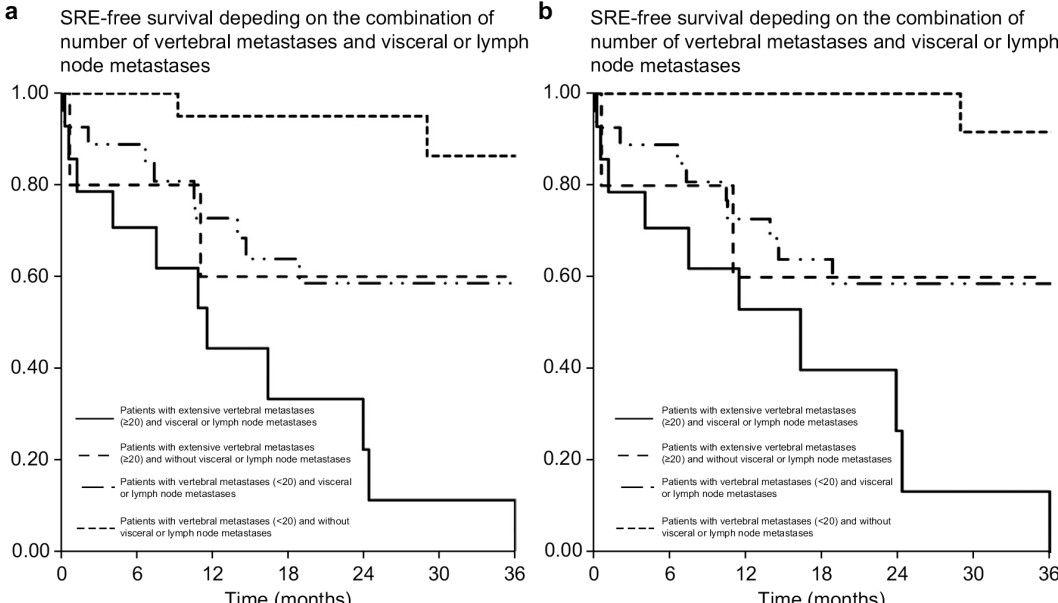

**Fig 4. SRE-free survival and spinal SRE-free survival based on the combination of vertebral metastases count and visceral or lymph node metastases.** Patients with extensive vertebral metastases (≥20) and visceral or lymph node metastases, with extensive vertebral metastases (≥20) and without visceral or lymph node metastases, with fewer vertebral metastases (<20) and visceral or lymph node metastases, and with fewer vertebral metastases (<20) and without visceral or lymph node metastases. SRE, skeletal-related event.

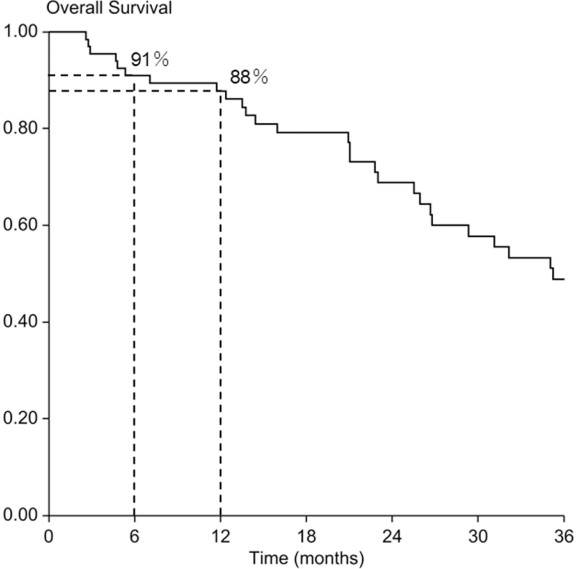

**Fig 5. Overall survival.** Overall survival rates were 91% and 88% at 6 and 12 months, respectively.

cord SREs typically coincided with disease advancement and later stages. These findings suggest that extensive vertebral involvement and metastases beyond the bone may serve as crucial indicators for predicting SREs.

The research demonstrates a positive correlation between the number of vertebral metastases and the incidence of SREs, particularly those affecting the spine. As cancer progresses, metastases in the vertebrae tend to multiply, with the spine being the predominant location for bone metastases [9]. A higher number of vertebral metastases typically signifies more advanced disease stages and consequently elevates the probability of experiencing SREs. While a connection between vertebral metastasis count and SRE occurrence has been observed in breast cancer patients, the precise nature of this relationship remains unclear for other cancer types. We found that patients with visceral or lymph node metastases developed their first SRE significantly earlier than those without visceral metastases. Ruchalski et al. reported that patients with visceral metastases before treatment were 1.5 times more likely to have disease progression compared to patients with metastatic CRPC [10]. Visceral metastases tend to occur later in the disease course, and patients with visceral involvement have a worse prognosis compared to those with only bone metastases in patients with CRPC [11–14]. Early detection of patients prone to SREs is crucial, as it enables clinicians to implement targeted surveillance for bone metastases, potentially averting severe health complications [2,7,8]. This proactive approach allows clinicians to focus their attention on high-risk patients, ensuring timely interventions and improved patient outcomes.

Appropriate collaboration between orthopedic surgeons and radiologists can help maintain patients' ability to perform daily activities by preventing pathological fractures or paraplegia caused by MSCC [9,15,16]. We have developed a predictive model of SREs using a combination of risk factors for more effective surveillance. Patients with extensive vertebral metastases (≥20) and visceral or lymph node metastases developed the first SRE and the first spinal SRE significantly earlier than did the other groups: the incidence of the first SRE and first spinal SRE at 12 months was 56% and 47%, respectively. Close monitoring for bone metastases is recommended in patients with visceral, lymph node, or extensive vertebral metastases (≥20).

The study has some limitations. First, this retrospective study was conducted at a single institution and involved a relatively small patient cohort (n = 68). Further research encompassing multi-center studies with larger cohorts is needed to validate the generalizability of the proposed model and refine risk subgroup thresholds. Additionally, although our findings identified a subset of CRPC patients with bone metastases at elevated risk for SREs, we were unable to establish optimal screening frequencies or methodologies for detecting bone metastases in these individuals. Early detection of hidden MSCC through spinal MRI, before neurological symptoms appear, can facilitate prompt intervention and potentially reduce the incidence of paralysis. Research on CRPC patients with bone metastases has shown a correlation between the extent of spinal metastases and clinically undetectable MSCC. Consequently, spinal MRI screening is recommended for patients at high risk of SREs [17]. Patients with CRPC who have extensive vertebral metastases (≥20) along with visceral or lymph node metastases may benefit from spinal MRI to detect and treat hidden MSCC early. However, no guidelines currently recommend specific imaging modalities [18–20]. Prospective randomized trials involving a significant patient cohort are required to validate the efficacy of early MSCC identification and treatment in high-risk groups and to determine the optimal frequency and utility of MRI screening. In conclusion, CRPC patients with visceral and lymph node metastases and extensive vertebral metastases (≥20) may be at high risk for SREs. Regular monitoring of these patients is crucial for early detection and management of bone metastases, thereby reducing complications and preserving their well-being. However, in this study, the number of spinal metastases was categorized into groups of 20, and there was an insufficient number of cases to allow accurate determination of the number of metastases. Additional research is necessary to identify predictors of SREs, and these findings should be confirmed through validation studies using independent datasets. Although it was not possible in the present study, integrating genomic/transcriptomic data from metastatic biopsies and exploring molecular correlates of SRE risk could provide new insights.

 

## Acknowledgments

We would like to thank Editage (www.editage.jp) for English language editing.

## Author contributions

**Data curation:** Tomohiro Inoue, Shota Takihira.

**Formal analysis:** Yoshiteru Akezaki.

**Investigation:** Haruyoshi Katayama, Takuto Itano.

**Project administration:** Eiji Nakata.

**Resources:** Shinsuke Sugihara.

**Supervision:** Toshifumi Ozaki.

**Visualization:** Ryuichi Nakahara.

**Writing – original draft:** Masanori Hamada.

**Writing – review & editing:** Eiji Nakata, Toshifumi Ozaki.

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
