## [Decision Letter · Decision Letter 0]

Dear Dr.  Nakata,

Thank you for submitting your manuscript to PLOS ONE. After careful consideration, we feel that it has merit but does not fully meet PLOS ONE’s publication criteria as it currently stands. Therefore, we invite you to submit a revised version of the manuscript that addresses the points raised during the review process.

We look forward to receiving your revised manuscript.

Kind regards,

Khalid Said Mohammad

Academic Editor

PLOS ONE

Journal Requirements:

2. In this instance it seems there may be acceptable restrictions in place that prevent the public sharing of your minimal data. However, in line with our goal of ensuring long-term data availability to all interested researchers, PLOS’ Data Policy states that authors cannot be the sole named individuals responsible for ensuring data access (http://journals.plos.org/plosone/s/data-availability#loc-acceptable-data-sharing-methods).

Additional Editor Comments:

Based on the reviewer comments, "see below." The manuscript will require major revision before considered for publication

Reviewers' comments:

Reviewer's Responses to Questions

**Comments to the Author**

1. Is the manuscript technically sound, and do the data support the conclusions?

Reviewer #1: Partly

Reviewer #2: Partly

2. Has the statistical analysis been performed appropriately and rigorously?

Reviewer #1: Yes

Reviewer #2: Yes

3. Have the authors made all data underlying the findings in their manuscript fully available?

Reviewer #1: Yes

Reviewer #2: Yes

4. Is the manuscript presented in an intelligible fashion and written in standard English?

Reviewer #1: Yes

Reviewer #2: Yes

Reviewer #1: Masanori Hamada et al developed a risk stratification model to predict skeletal-related events (SREs) in patients with bone metastases from castration-resistant prostate cancer (CRPC), identifying extensive vertebral metastases and visceral/lymph node metastases as key risk factors through multivariate analysis. Patients with the highest-risk group (extensive vertebral plus additional metastases) showed significantly earlier onset of SREs and spinal SREs, reaching 56% and 47% incidence at 12 months, respectively. The findings emphasize close monitoring for high-risk subgroups to prevent SREs and guide personalized management in CRPC patients with bone metastases, ensuring timely interventions and improved patient outcomes.

The manuscript provides valuable insights into SREs of CRPC. I will give some comments as follows.

1. The sample size (n=68) was small. While the cohort provided preliminary insights into risk stratification for SREs, future multi-center studies with larger cohorts are needed to validate the generalizability of the proposed model and refine risk subgroup thresholds.

2. Fig. 5 should be revised to improve clarity, including adding detailed information and expanding the figure legend to explicitly define key findings.

3. I will propose validating the findings using datasets specific to advanced CRPC or integrating genomic/transcriptomic data from metastatic biopsies to explore molecular correlates of SRE risk.

Reviewer #2: Hamada et al present a study that looks to stratify castrate-resistant prostate cancer patients with bone metastases based on their likelihood of developing skeletal-related events (SRE). The manuscript is clearly laid out and well written. However, I am still unsure of the novelty, impact, and in parts, the experimental design. For instance, a central finding of the manuscript is that patients with a higher count of vertebrae metastases have a higher risk of SRE. Unless I am missing a key detail, this seems intuitive and borderline obvious… In other words, more cancer = more risk of an event… As a result, it is not clear how the findings from this manuscript will add to our clinical understanding or management of the disease. Personally, I think that this is a key point that needs to be addressed before this work can be published.

Minor points:

Why was 20 picked as a cut off for more or less vertebrae metastases?

The patient cohort is 10 – 20 years old and prior to current best practices. Can the authors comment on how this time lag is expected to impact their model?

Why was no comparison performed between zoledronic acid and denosumab? I would have thought that this would be a key question given that both agents are used to try and prevent SREs.

**Do you want your identity to be public for this peer review?** For information about this choice, including consent withdrawal, please see our Privacy Policy

Reviewer #1: **Yes: ** Xiaofeng Ding

Reviewer #2: No

---

## [Author Response · Author response to Decision Letter 1]

16 Jun 2025

Response letter

June 1, 2025

Khalid Sail Mohammad

Academic Editor

PLoS One

Dear Editor:

We are pleased to submit our revised manuscript, (ID: PONE-D-24-56347), titled “Risk stratification for the prediction of skeletal-related events in patients with castration-resistant prostate cancer with bone metastases.” First, we would like to express our deepest gratitude to the reviewers and you for the constructive and insightful comments.

The feedback was invaluable in enhancing the quality and clarity of our research.

We have carefully considered each comment and suggestion made by the reviewers and made corresponding revisions to the manuscript, indicated as yellow-highlighted text. Below, we address each comment point-by-point and outline the changes made in the revised manuscript:

Reviewer #1: Masanori Hamada et al developed a risk stratification model to predict skeletal-related events (SREs) in patients with bone metastases from castration-resistant prostate cancer (CRPC), identifying extensive vertebral metastases and visceral/lymph node metastases as key risk factors through multivariate analysis. Patients with the highest-risk group (extensive vertebral plus additional metastases) showed significantly earlier onset of SREs and spinal SREs, reaching 56% and 47% incidence at 12 months, respectively. The findings emphasize close monitoring for high-risk subgroups to prevent SREs and guide personalized management in CRPC patients with bone metastases, ensuring timely interventions and improved patient outcomes.

The manuscript provides valuable insights into SREs of CRPC. I will give some comments as follows.

1. The sample size (n=68) was small. While the cohort provided preliminary insights into risk stratification for SREs, future multi-center studies with larger cohorts are needed to validate the generalizability of the proposed model and refine risk subgroup thresholds.

Response: Thank you for your comments. Indeed, the small sample size was a limitation, and we agree that multi-center studies with larger cohorts are necessary. Accordingly, we have added these points in the limitations sections of the Discussion, as follows.

“The study has some limitations. First, this retrospective study was conducted at a single institution and involved a relatively small patient cohort (n=68). Further research encompassing multi-center studies with larger cohorts is needed to validate the generalizability of the proposed model and refine risk subgroup thresholds”(Line 246-249)

2. Fig. 5 should be revised to improve clarity, including adding detailed information and expanding the figure legend to explicitly define key findings.

Response: Thank you for your valuable suggestions. Accordingly, we have corrected the legend for Figure 5 to ensure the numbers are accurate (Lines 209-210).

We have also added more detailed information to Figure 5 and revised it to make it easier to understand.

3. I will propose validating the findings using datasets specific to advanced CRPC or integrating genomic/transcriptomic data from metastatic biopsies to explore molecular correlates of SRE risk.

Response: Indeed, the results of our study need to be validated using a dataset specific to progressive CRPC. In the original paper, we had stated "Additional research is necessary to identify predictors of SREs, and these findings should be confirmed through validation studies using independent datasets.”(Line 264-265)

To further emphasize this point, we have added the following test to the Conclusion: “Although it was not possible in the present study, integrating genomic/transcriptomic data from metastatic biopsies and exploring molecular correlates of SRE risk could provide new insights.” (Line 270-272)

Reviewer #2: Hamada et al present a study that looks to stratify castrate-resistant prostate cancer patients with bone metastases based on their likelihood of developing skeletal-related events (SRE). The manuscript is clearly laid out and well written. However, I am still unsure of the novelty, impact, and in parts, the experimental design. For instance, a central finding of the manuscript is that patients with a higher count of vertebrae metastases have a higher risk of SRE. Unless I am missing a key detail, this seems intuitive and borderline obvious… In other words, more cancer = more risk of an event… As a result, it is not clear how the findings from this manuscript will add to our clinical understanding or management of the disease. Personally, I think that this is a key point that needs to be addressed before this work can be published.

Response: Thank you for your insightful comments. We aimed to develop a metric that could detect hidden metastatic spinal cord compression (MSCC) in patients with castration-resistant prostate cancer (CRPC) and bone metastases. Early detection of hidden MSCC by spinal cord magnetic resonance imaging before neurological symptoms develop could enable rapid intervention and reduce the risk of paralysis. Some studies have suggested that patients with CRPC with visceral, lymph node, or extensive vertebral metastases (more than 20) may be at a higher risk for SRE. However, further studies are needed to identify the predictors of SRE. These findings should be confirmed by validation studies using independent data sets.

Minor points:

Why was 20 picked as a cut off for more or less vertebrae metastases?

Response: Thank you for this question. It is difficult to provide a clear rationale for dividing the number of spinal metastases by 20. In this study, most patients had 10 or fewer spinal metastases and the number of metastases throughout the spine was 24. Of the 60 patients who had spinal metastases at the last observation, 27 had metastases throughout the spine; 29 patients had 20 or more metastases, while 26 had 10 or fewer metastases. Therefore, the number of spinal metastases was set at 20. However, we believe that further research is needed to create a clear rationale; therefore, we have added this following information to the revised manuscript:

“However, in this study, the number of spinal metastases was categorized into groups of 20, and there was an insufficient number of cases to allow accurate determination of the number of metastases.” (Line 266-268)

The patient cohort is 10 – 20 years old and prior to current best practices. Can the authors comment on how this time lag is expected to impact their model?

Response: We appreciate your question. The cohort in this study reflects treatment patterns, survival outcomes, and disease progression characteristics from a time when treatment options were much more limited and treatment criteria were different. Early detection of hidden metastatic spinal cord compression (MSCC) by spinal cord magnetic resonance imaging before the onset of neurological symptoms would allow for rapid intervention and potentially reduce the incidence of paralysis. The aim of this study was to create an index to detect hidden MSCC in patients with castration-resistant prostate cancer patients with bone metastases.

Why was no comparison performed between zoledronic acid and denosumab? I would have thought that this would be a key question given that both agents are used to try and prevent SREs.

Response: Thank you for your comment. We recognize that comparing zoledronic acid and denosumab is an important clinical issue because healthcare providers widely use both drugs to prevent bone-related events in patients with bone metastases. Several large randomized controlled trials and meta-analyses have been conducted to compare the two drugs directly, showing that denosumab is superior to or equivalent with zoledronic acid in delaying or suppressing skeletal-related events (SREs).

We aimed to develop a metric that could detect hidden metastatic spinal cord compression (MSCC) in patients with castration-resistant prostate cancer (CRPC) and bone metastases. The primary objective of this study was not to make a direct comparison between the two drugs. However, we acknowledge the need for more detailed comparisons and subgroup analyses and plan to conduct additional studies with larger patient populations in the future.

In conclusion, we believe that these revisions have helped significantly improve our manuscript, making it a valuable contribution to “PLoS One”. We appreciate the opportunity to revise our work and thank the reviewers for their essential role in this process.

Thank you for considering our revised manuscript for publication in “PLoS One”. We look forward to your decision.

Sincerely,

Masanori Hamada,

Division of Physical Medicine and Rehabilitation, Okayama University Hospital, 2-5-1, Shikata-cho, Okayama City, Okayama 700-8558, Japan

E-mail: pjvy4uc1@okayama-u.ac.jp (EN)

Eiji Nakata

Department of Orthopedic Surgery, Okayama University Hospital, 2-5-1, Shikata-cho, Okayama City, Okayama 700-8558, Japan

E-mail: eijinakata8522@yahoo.co.jp (EN)

---

## [Decision Letter · Decision Letter 1]

Risk stratification for the prediction of skeletal-related events in patients with castration-resistant prostate cancer with bone metastases

PONE-D-24-56347R1

Dear Dr. Nakata,

We’re pleased to inform you that your manuscript has been judged scientifically suitable for publication and will be formally accepted for publication once it meets all outstanding technical requirements.

Kind regards,

Khalid Said Mohammad

Academic Editor

PLOS ONE

Additional Editor Comments (optional):

Reviewers' comments:

Reviewer's Responses to Questions

**Comments to the Author**

Reviewer #1: All comments have been addressed

Reviewer #2: (No Response)

2. Is the manuscript technically sound, and do the data support the conclusions?

Reviewer #1: Yes

Reviewer #2: Yes

3. Has the statistical analysis been performed appropriately and rigorously?

Reviewer #1: Yes

Reviewer #2: Yes

4. Have the authors made all data underlying the findings in their manuscript fully available?

Reviewer #1: Yes

Reviewer #2: Yes

5. Is the manuscript presented in an intelligible fashion and written in standard English?

Reviewer #1: Yes

Reviewer #2: Yes

Reviewer #1: The authors have addressed all concerns raised during the review process, and I agree that the current version of the paper is accepted.

Reviewer #2: This manuscript is greatly improved, and I am satisfied with the authors responses. I believe that this is now ready to be published in PLOS One.

**Do you want your identity to be public for this peer review?** For information about this choice, including consent withdrawal, please see our Privacy Policy

Reviewer #1: **Yes: ** Xiaofeng Ding

Reviewer #2: No

---

## [Editor Report · Acceptance letter]

PONE-D-24-56347R1

PLOS ONE

Dear Dr. Nakata,

I'm pleased to inform you that your manuscript has been deemed suitable for publication in PLOS ONE. Congratulations! Your manuscript is now being handed over to our production team.

Kind regards,

on behalf of

Dr. Khalid Said Mohammad

Academic Editor

PLOS ONE